# Dynamic Response Analysis of Submarines Based on FEM-ALE Coupling Method in Floating Ice Conditions

Zhongyu Chen [1], Weidong Zhao [1,2,*], Zhanyang Chen [1], Guoqing Feng [2], Huilong Ren [2] and Hongbin Gui [1]

1   School of Ocean Engineering, Harbin Institute of Technology, Weihai 264209, China
2   College of Shipbuilding Engineering, Harbin Engineering University, Harbin 150009, China
*   Correspondence: weidong.zhao@hit.edu.cn

**Abstract:** To address global challenges, research on the safety of polar navigation is indispensable. However, most of studies focus on traditional surface vessels, with few research studies on submarine. The dynamic response of submarine during surface navigation in floating ice channels under special conditions is studied in this work. Firstly, a model of the submarine incorporating an intact internal frame was established. Subsequently, the FEM-ALE coupled method was employed to simulate the structure-ice interaction, and the obtained results was verified by the Colbourne method. Then, the parametric study (navigation speed, ice thickness, and floating ice size) were analyzed from the perspectives of ice resistance, stress and plastic strain. Finally, an empirical equation suitable for the interaction between submarine and floating ice during surface navigation is improved based on the Colbourne method.

**Keywords:** structure-ice collision; FEM-ALE; submarines; ice resistance; floating ice condition

## 1. Introduction

With the increasing degree of globalization and the ever-growing energy shortage problem, the Arctic region harbors paramount developmental significance [1]. The Arctic region has a transcontinental route linking Asia and Europe, and it contains a large amount of untapped resources [2]. However, numerous natural obstacles, such as floating ice, layer ice, and ridges, as well as extreme environmental conditions, pose a serious threat to the safety of polar navigation. Therefore, safety assessments of structures such as icebreakers, commercial vessels, and submarines that navigate in the Arctic region are particularly important [3]. Current researches mainly focuses on traditional surface vessels, such as icebreakers and commercial vessels, with limited research on submarines.

The dynamic response during ship-ice interaction process is affected by the shape of the ship, structural strength, and ice properties [4]. Analytical methods that are conventional have difficulty comprehensively considering multiple factors. Therefore, researchers commonly use empirical equation methods, simulation methods, and experimental methods. The experimental method can be divided into full-scale ship tests and model tests, which yield more accurate results but are costly and time consuming. Empirical equations are derived from the summary and integration of data from considerable of tests, and often apply only to a certain type of operating condition. A relatively lower cost is associated with the simulation method, rendering it applicable to a majority of operational conditions [5].

In the early nineteenth century, experimental methods were primarily employed for the accumulation of a significant volume of crucial data, from which empirical equations were derived. The earliest ice resistance equation, presented by Runeberg [6], took into consideration the influences of frictional forces and bow trim angles. Ship design parameters were initially considered by Shimanskii [7], while Kashteljan [8] pioneered the partitioning of ice resistance into breaking resistance, overturning and submergence resistance, and damaged floating ice resistance. Jones [9] introduced semi-empirical methods into the

study of continuous ice breaking. Based on the aforementioned research and the accumulation of a substantial amount of data, numerous empirical equations with notable effectiveness were developed. For level ice conditions, methods such as the Lindqvist [10] approach, which incorporated considerations of friction and ship hull geometry, as well as the Riska [11] method derived from the modified Lindqvist approach, had been established. Conversely, for floating ice conditions, the relevant empirical equations were predominantly compiled and summarized from experimental data. Examples include the Bronnikov method [12] based on ship tests and the Mellor method based on the Mohr-Coulomb failure criterion [13].

In the experimental aspect, Daley [14] conducted observations on the failure process of the ice layer through full-scale and model-scale experiments and proposed a contact model for the ice layer's edge. Jeong [15] conducted ice tank experiments with square-shaped floating ice in three different channel widths and analyzed the impact of ice size on resistance, while proposing a rapid method for calculating ice concentration. Kim [16] conducted model-scale ice resistance experiments in a simulated ice environment using triangular ice elements made of paraffin instead of real ice. Several operating conditions were set based on different velocities, ice concentration, and ship hull waterline entry angles. The experimental results were compared with numerical simulation results. Jeong [17] and others conducted model-scale ship tests in a frozen ice region at the MOERI ice tank, resulting in the prediction of planar ice resistance for various thicknesses and bending strengths.

With the advancement of computer technology and numerical simulation methods, researchers have begun to employ simulation techniques to study the physical processes and mechanical characteristics of ice collisions. Among these, FEM, SPH, DEM, and their combined usage are the most common research approaches employed today.

The application of the FEM to simulate ice loads was pioneered by Määttänen and Hoikkanen in 1990, while Evgin et al. subsequently successfully utilized the DEM for ice-breaking simulations [18]. Munjiza [19] proposed the FEM-DEM method, which integrated the advantages of both approaches, using FEM to simulate ice fracture and DEM to simulate ice accumulation. Huang et al. [20] based on the combination of CFD and DEM methods, investigated fragmented ice channels and discovered a linear relationship between the thickness and diameter of floating ice and ice resistance. Based on the ice shell collision mechanism and fundamental icebreaking characteristics, Karl [21] introduced a simplified numerical model to predict the ice impact forces acting on the vessel under horizontal ice conditions. The model addresses two critical failure modes, namely local crushing and flexural fracture. Andrei [22] constructed a CFD numerical model based on the DEM-BEM theory, suitable for simulating the interaction between floating structures and fragmented ice. The potential flow theory was employed to predict the flow field around the ship hull and the surrounding fragmented ice. Finally, the computational results were compared with the ice tank test results.

Zhang [23] performed numerical simulations of the collision between a vertical cylindrical structure and layered ice by utilizing LS-DYNA finite element software. Two different thicknesses of cohesive element ice models were constructed, and the S-ALE fluid–structure coupling method was employed. Song, Kim, and others [24] utilized the ALE method within the LS-DYNA finite element software to simulate the mutual interaction between floating ice and marine structures. They accounted for the fluid–structure interaction problem and validated the correctness of the ALE method and the coupling algorithm in the interaction between fluid, ice, and structures.

Apart from those, Su [25] incorporated numerical models to investigate the overall and local ice loads on ship structures. Through two case studies, simulated ice loads on the vessel were examined, analyzing ship performance, statistical framework loads induced by ice, and the spatial distribution of ice loads around the ship. A comparison with field measurements was conducted as well. Chai [26] applied probability methods and models to seek the correlation between ice-induced load statistics and major ice conditions in the field of ship and ocean engineering. Zhao [27] performed the probability-based fatigue

damage assessment of vessels traversing horizontal ice fields. A novel procedure utilizing numerical simulation had been developed for the evaluation of fatigue damage, and the process was demonstrated through a hypothetical scenario involving the icebreaking vessel Snow Dragon 2. The sensitivity of the procedure to key analysis parameters, such as sample size and initial crack size, was also taken into consideration by Zhao [28]. The impact of low temperature on the computational results was analyzed as well.

The icebreaking research mentioned above primarily focuses on surface vessels, with limited experimental studies and insufficient available data conducted on submarines, thus few related empirical equations have been proposed for the ice resistance of submarines. However, they may encounter collisions with small-scale floating ice when they need to surface during certain special emergency tasks. This paper aims to investigate the dynamic response of submarines navigating through the surface of floating ice-field using the FEM-ALE coupling approach. The effects of navigation speed, ice thickness, and floating ice size on the polar navigation safety of submarines are studied. Special emphasis is placed on the ice resistance force-time history, as well as the stress–strain distribution on both the surface and interior of the structures, to propose a reference condition for safe navigation. The novelty of this study lies in the fact that the submarines are not modeled as empty shells, but as fully integrated bow structures with internal plates, frames, and trusses. Based on the numerical results, critical conditions and empirical equations are proposed.

## 2. Theoretical Background

### 2.1. The Theory of Ship-Ice Interaction

The collision between ships and ice involves a significant amount of non-linear behaviors, such as material non-linearity, geometric non-linearity, and contact non-linearity. Material non-linearity refers to the complex stress–strain relationship of sea ice material, which is influenced by multiple factors. Geometric non-linearity refers to the large deformation of sea ice, which is not suitable for the small deformation principle. Contact non-linearity refers to the unstable contact between two substances, resulting in a large number of non-linear changes. In this work, comprehensive consideration is given to the three non-linearities, where both the sea ice and structure are simplified as elastic-plastic materials, and the contact is computed using an automatic face-to-face contact algorithm. In terms of the algorithm, the simulation calculation of the ship collision process essentially involves the numerical solution of collision equations, which is accomplished using the explicit central difference method [29].

When the results of $0, \ldots, t_n$, time steps are known:

$$M\ddot{U}(t_n) = P(t_n) - F^{\text{int}}(t_n) + H(t_n) - C\dot{U}(t_n) \tag{1}$$

where: $P(t)$—external force vector;
$F^{\text{int}}(t_n)$—internal force vector;
$F^{\text{int}}(t_n) = \sum \int_{\Omega} B^T \sigma_n d\Omega + F^{contact}$, Nodal element internal force;
$H(t_n)$—hourglass resistance.
Inverting the equation, it can get:

$$\ddot{U}(t_n) = M^{-1}\left[P(t_n) - F^{\text{int}}(t_n) + H(t_n) - C\dot{U}(t_n)\right] \tag{2}$$

According to the definition of finite difference method, the displacement, velocity, and acceleration at time $t_{n+1}$ can be solved by the following equation:

$$\dot{U}(t_{n+\frac{1}{2}}) = \dot{U}(t_{n-\frac{1}{2}}) + \ddot{U}(t_n)\Delta t_n \tag{3}$$

$$U(t_{n+1}) = U(t_n) + \dot{U}(t_{n+\frac{1}{2}})\Delta t_{n+\frac{1}{2}} \tag{4}$$

where: $\Delta t_{n+\frac{1}{2}} = \frac{\Delta t_n + \Delta t_{n+1}}{2}$, $\Delta t_{n-\frac{1}{2}} = \frac{\Delta t_n + \Delta t_{n-1}}{2}$.

After a series of calculations, the mass influence coefficients are transformed into a diagonal matrix, achieve decoupling, and enable independent computations. In this calculation process, the influence of the stiffness matrix is assumed to be neglected, and only the central single-point integration is considered, significantly reducing the computational steps and decreasing computation time.

### 2.2. The Material Model

Sea ice is a typical nonlinear material whose physical properties are influenced by various factors, including load strain, temperature, ice age, and salinity. Describing the constitutive model of sea ice accurately in simulation calculations is challenging, and existing research often replaces sea ice materials with ice load models.

At present, there is no perfect ice material model due to the complex physical properties and microstructure of ice. Based on experimental and simulation results, researcher [30] have proposed several reliable ice load models, including elasto-plastic models based on plasticity theory, viscoplastic models, improved models, anisotropic failure models, and visco-elastic-plastic models based on the particle flow theory of sea ice, as well as crushable foam models.

For the elastic-plastic model [31], it is assumed that the material undergoes only elastic deformation when the stress is relatively low. However, when the stress exceeds the yield stress, the stress–strain curve deviates from the elastic stage and becomes a sloping line. Once the yield stage is reached, the material starts exhibiting irreversible plastic deformation, and as the stress continues to increase, the material may fail and undergo damage.

Based on the existing experiments, the stress-strain curve of ice floe is shown in red line in Figure 1. However, elastoplastic models are often used instead of actual ice floes in the simulation of structure-ice interaction. The curve is approximated as two straight lines, the former part is the elastic stage, and the latter part is the plastic stage. The failure modes of sea ice can be simplified into four types: local fragmentation caused by compression failure, instability caused by buckling failure, cracks generated by shear failure, and fractures caused by bending failure. This study focuses on maximum floating ice sizes of only 6 m × 6 m, which are relatively small and rarely experience the latter three failure modes. Typically, only compression failure occurs, specifically in the parts directly in contact with the structure. In this work, the elastic-plastic model is utilized, which effectively captures most properties of the floating ice and enables failure simulation through mesh removal.

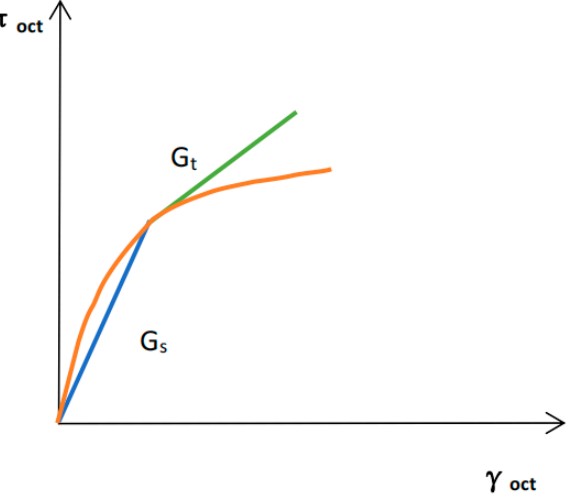

**Figure 1.** Simplified schematic diagram of elastic-plastic model [31].

### 2.3. FEM-ALE Coupling Method

Finite element method includes Lagrangian, Eulerian, and ALE methods [32]. In the Lagrangian method, element points are identical to material points, and the element is fixed to the object, moving along with the model nodes. In the Eulerian method, element points are spatial points, and the element is fixed in space, unaffected by the movement of material points. In the ALE description, a reference configuration independent of the actual and initial configurations is introduced, with element points as reference points. The motion of the element in space is arbitrary, independent of the Lagrangian and Eulerian coordinate systems, and can be chosen as needed. When the element motion velocity is reasonable, this method can accurately simulate object deformation and track object motion, making it suitable for modeling nonlinear and large deformation changes, particularly in fluid domains. As shown in Figure 2, A, B and C are grid movements of Lagrangian, Euler, and ALE methods respectively.

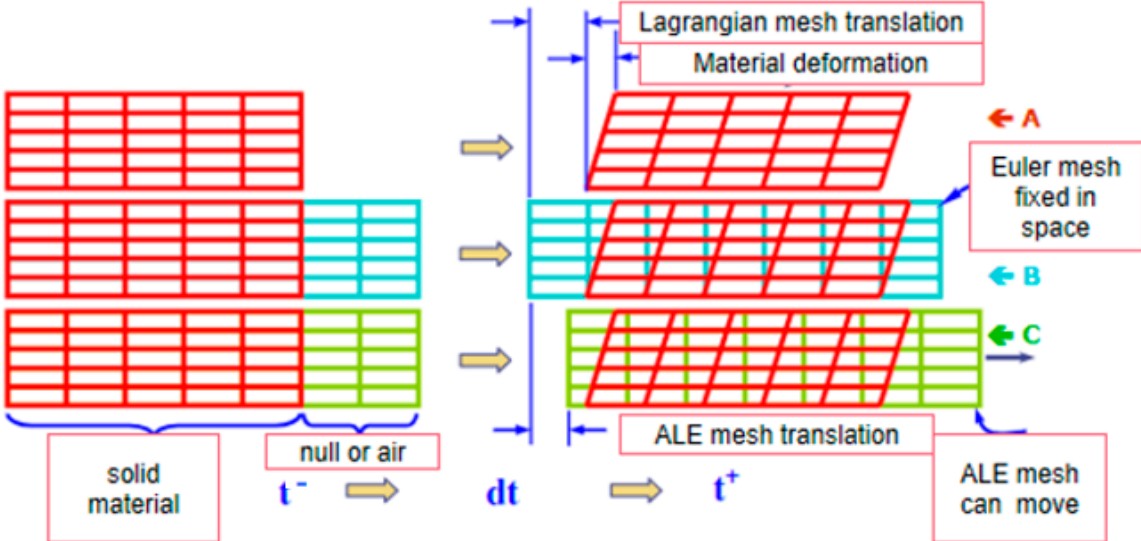

**Figure 2.** Schematic diagram of mesh movement.

In the traditional Eulerian conservation equations, the conventional terms have been replaced by relative velocity yielding the conservation equations in an arbitrary Lagrangian form, as expressed below [33]:

Mass conservation equation:

$$\frac{\partial \rho}{\partial t}|_x + c\cdot\nabla\rho = -\rho\nabla\cdot v \tag{5}$$

Momentum conservation equation:

$$\rho\left(\frac{\partial \rho}{\partial t}|_x + (c\cdot\nabla)v\right) = \nabla\cdot\sigma + \rho b \tag{6}$$

Energy conservation equation:

$$\rho\left(\frac{\partial \rho}{\partial t}|_x + c\cdot\nabla E\right) = \nabla\cdot(\sigma\cdot v) + v\cdot\rho b \tag{7}$$

Internal energy conservation equation:

$$\rho\left(\frac{\partial \rho}{\partial t}|_x + c\cdot\nabla e\right) = \sigma\cdot\nabla^S v \tag{8}$$

At each computational instant, two distinct stages are encompassed. In the first stage, material does not flow across the boundaries of the grid, and there is no material overflow throughout the entire calculation process, ensuring mass conservation. In the second stage, material flows across the edges of the elements, referred to as convection. The ALE method computes the transport quantities, internal energy, and momentum of the various physical properties as the material passes through the element boundaries. Unlike the first stage of the Lagrangian method, the second stage involves the generation of an independent motion in the grid, separate from the material. The grid's independent relative motion allows it to return to its original position or any other position that facilitates more accurate calculations. ALE does not support implicit time integration, dynamic relaxation, and contact. However, large deformation motion can be well described by ALE, which is applicable for fluid–structure interaction in fluid dynamics.

The main models in this work includes the submarine, floating ice, and fluid domains, with defined contacts between floating ice and the submarine, and coupling between each model and the fluid domain. Both floating ice and structures use an elastoplastic material model, representing contact between structures and floating ice as contact between elastic bodies. A penalty function algorithm is employed for contact, which checks if nodes penetrate the master surface at each time step and applies a large interface contact force at the penetrated locations. The magnitude of this force is proportional to the penetration depth and the stiffness of the master surface, ensuring high efficiency and applicability.

The collision between the ship and ice in this work is highly complex and involves erosion double-sided contact, where the structure and floating ice act as master and slave surfaces, respectively [34]. The fluid–structure interaction utilizes a penalty function algorithm, and the ALE moving mesh method is employed to calculate fluid motion. Due to the LS-DYNA explicit integration scheme utilizing the central difference method, the accuracy of calculations is affected by the hourglass deformation of quadrilateral meshes. For fluids, hourglass control based on a viscous equation is applied to suppress hourglass deformation, while for solids, hourglass control based on a stiffness equation is used to counteract hourglass deformation by deforming in the opposite direction.

### 2.4. Colbourne Method

The maximum ice resistance is often employed for evaluating structural strength, whereas the average ice resistance is generally utilized for calculating economic benefits. The validation of the maximum ice resistance results is carried out in this work. The Colbourne method [35], which treats the ice resistance as a unified entity by disregarding infrequent ice fracture occurrences, is adopted to validate the predicted maximum ice resistance based on the numerical simulation. This approximation aligns with the simulation settings in the computational software. According to Colbourne method the total resistance is divided into open water resistance and ice resistance, as expressed by the following equation:

$$R_{OW} = C_{OW}V^2$$
$$R_P = C_P\rho_i g B h_i V^2 C^n \tag{9}$$

where:

$C_{OW}$—the open water resistance coefficient,
$C_P$—the drag coefficient of ice floe,
$\rho_i$—the density of ice,
$h_i$—the ice thickness,
$g$—the acceleration of gravity,
$V$—the ship speed,
$C$—the concentration of ice floe.
The drag coefficient of floating ice is a dimensionless value, defined as follows:

$$C_P = \frac{R_P}{\rho_i g B h_i V^2 C^n} \tag{10}$$

where:

*B*—maximum beam for the structure

Similarly, the non-dimensionalization of navigation speed is also performed in the Colbourne method. It is considered that the Froude number is related to the ice concentration. The ice Froude number is defined as follows:

$$Fr_P = \frac{V}{\sqrt{gh_iC}} \tag{11}$$

Based on the Colbourne method, it is considered that there exists a linear relationship between the natural logarithm of *Cp* and the natural logarithm of *Fr$_p$*. However, in the case of this study, the structure is unique and lacks relevant experimental data for validation. Therefore, utilizing multiple simulated data obtained from our own simulations, the natural logarithm of *Cp* and the natural logarithm of *Fr$_p$* are calculated, and a corresponding curve can be plotted. The study aims to evaluate the accuracy by inversely verifying the error in the resistance results based on this curve.

## 3. The Interaction between Submarine and Floating Ice

### 3.1. The Numerical Setup

A typical submarine is chosen for the structure-ice interaction in this work. To simplify the computational process and ensure accuracy, only the bow part which includes an internally intact plate-frame structure, is utilized for collision analysis, The length of the bow of the structure is 9.2 m, the width is 7.8 m, the height above water is 2.1 m, and the height below water is 6.7 m.

The collision between the structure and the floating ice primarily occurs with the first ice floe. To reduce computational time while maintaining accuracy, the mesh size of the first ice floe is refined to 0.1 m × 0.1 m × 0.1 m, while the remaining ice floes have a mesh size of 0.2 m × 0.2 m × 0.2 m. The ice concentration is set at 50%, and several rectangular ice floes with cross-sectional dimensions of 6 m × 6 m, 4 m × 4 m, and 2 m × 2 m are considered. The structure-ice interaction model and the investigated operating conditions are presented in Figure 3 and Table 1, respectively.

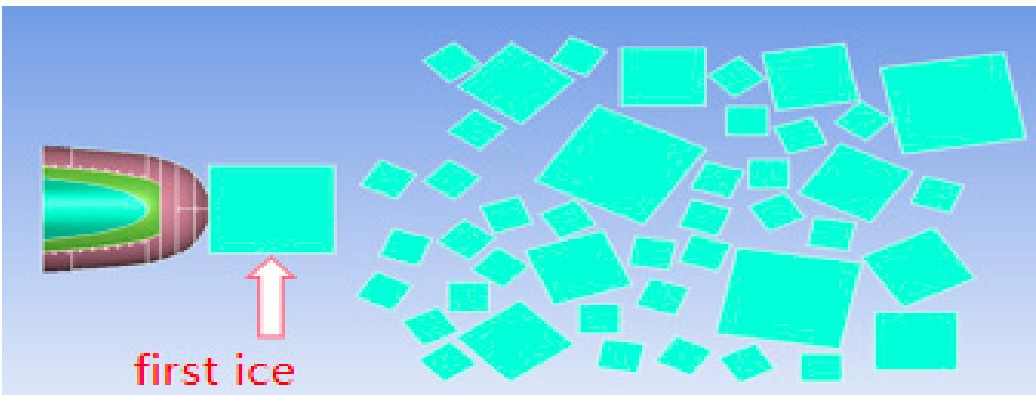

**Figure 3.** The numerical model for structure-ice interaction.

**Table 1.** Working condition parameter.

| No. | Speed (kn) | Ice Thickness (m) | The Size of the First Floating Ice (m × m) |
|-----|------------|-------------------|--------------------------------------------|
| 1 | 6 | 1.2 | 6 × 6 |
| 2 | 4 | 1.2 | 6 × 6 |
| 3 | 2 | 1.2 | 6 × 6 |

| No. | Speed (kn) | Ice Thickness (m) | The Size of the First Floating Ice (m × m) |
|-----|-----------|-------------------|--------------------------------------------|
| 4 | 1 | 1.2 | 6 × 6 |
| 5 | 6 | 1.0 | 6 × 6 |
| 6 | 6 | 0.8 | 6 × 6 |
| 7 | 6 | 0.5 | 6 × 6 |
| 8 | 6 | 1.2 | 4 × 4 |
| 9 | 6 | 1.2 | 2 × 2 |

*3.2. The Material Parameters*

3.2.1. The Fluid Domain

The fluid domain (sea water and air) adopts the null hydrodynamic material type in LS DYNA. The related parameters are given in Table 2.

**Table 2.** The material parameters for the fluid domain.

| | Density (kg/m$^3$) | Cut off Pressure (Pa) | Viscosity Coefficient (Pa·s) |
|-----------|-------------------|----------------------|------------------------------|
| Sea water | 1.03 | −1000 | $1.79 \times 10^{-3}$ |
| Air | 1018 | −100 | $1.75 \times 10^{-5}$ |

The effects of seawater on the behavior of the ship and ice can be considered by application of the ALE method. In LS-DYNA, the computation of fluids necessitates the definition of materials and EOS. An EOS such as the Gruneisen model is suggested to simulate water in the current FEM solver. The Gruneisen EOS with cubic shock velocity-particle velocity defines the pressure for a compressed material.

$$P = \frac{\rho_0 C^2 \mu \left[1 + \left(1 - \frac{\gamma_0}{2}\right) - \frac{\alpha}{2}\mu^2\right]}{\left[1 - (S_1 - 1)\mu - S_2 \frac{\mu^2}{\mu+1} - S_3 \frac{\mu^3}{(\mu+1)^2}\right]^2} + (\gamma_0 + \alpha\mu)E_0 \tag{12}$$

where:

$E$—the internal energy per initial volume,

$C$—the intercept of the $\mu_s - \mu_p$ curve (speed of sound in water),

$S_1$, $S_2$ and $S_3$—the coefficients of the slope of the $\mu_s - \mu_p$ curve,

$\gamma_0$—the Gruneisen gamma parameter,

$\alpha$—the first-order volume correction to $\gamma_0$.

The seawater characteristics based on the relevant parameters are given in Table 3.

**Table 3.** The EOS parameters for sea water.

| | C | S1 | S2 | GAMAO | E$_0$ (J) | V$_0$ (m$^3$) |
|-----------|----------|-------|--------|-------|---------------------|----------------|
| Sea water | 1480.000 | 1.920 | −0.096 | 0.350 | $2.895 \times 10^5$ | 1.000 |

The air domain adopts the linear equation EOS, and the specific equation is shown in Equation (13). The related parameters are given in Table 4.

$$P = C_0 + C_1\mu + C_2\mu^2 + C_3\mu^3 + \left(C_4 + C_5\mu + C_6\mu^2\right)E \tag{13}$$

**Table 4.** Table The EOS parameters for air.

|  | **C0** | **C1** | **C4** | **C5** | **E$_0$ (J)** | **V$_0$ (m$^3$)** |
|---|---|---|---|---|---|---|
| Air | 0.000 | 0.000 | 0.400 | 0.400 | $2.530 \times 10^5$ | 1.000 |

### 3.2.2. The Material Parameters for Submarines

In this work, an elastic-plastic model more suitable for high-strength steel is adopted which defines the structural deformation during the collision process of structures. Related material parameters are given in Table 5.

**Table 5.** The material parameters for submarines.

| **Density (kg/m$^3$)** | **Elastic Modulus (Pa)** | **Possion's Ratio** | **Yield Strength (Pa)** | **Tangent Modulus (Pa)** | **SRC** | **SRP** | **Plastic Failure Strain** |
|---|---|---|---|---|---|---|---|
| 7850.0000 | $2.060 \times 10^{11}$ | 0.300 | $3.900 \times 10^8$ | $1.180 \times 10^9$ | 3200 | 5.00 | 0.28 |

Where: SRC—Strain rate parameter *C*; SRP—Strain rate parameter *p*.

### 3.2.3. The Material Parameters for Floating Ice

The ice material employed in this work utilizes the previously mentioned isotropic elastic failure model. This model exhibits elastic deformation when subjected to low stress levels. However, plastic deformation occurs when the stress exceeds the yield strength. In the event that the plastic strain surpasses 0.002, the corresponding mesh elements are automatically deleted due to failure. The related parameters of the model are given in Table 6.

**Table 6.** The material parameters for floating ice.

| **Density (kg/m$^3$)** | **Tangent Modulus (Pa)** | **Yield Strength (Pa)** | **Hardening Modulus (Pa)** | **Plastic Failure Strain** | **Failure Pressure (Pa)** | **Bulk Modulus (Pa)** |
|---|---|---|---|---|---|---|
| 920.00 | $2.20 \times 10^9$ | $2.12 \times 10^6$ | $4.26 \times 10^9$ | 0.002 | $-4.00 \times 10^6$ | $5.26 \times 10^9$ |

### *3.3. The Verification for Numerical Model*

#### 3.3.1. The Verification for Pressure Gradient with Single Floating Ice

Before the numerical simulation of structure-ice interaction, it is necessary to validate the stability of the fluid pressure and the ability of a single ice floe to float stably. The air pressure validation is conducted by examining the variations in air pressure gradient and the temporal changes in air pressure at monitored nodes, as shown in Figure 4.

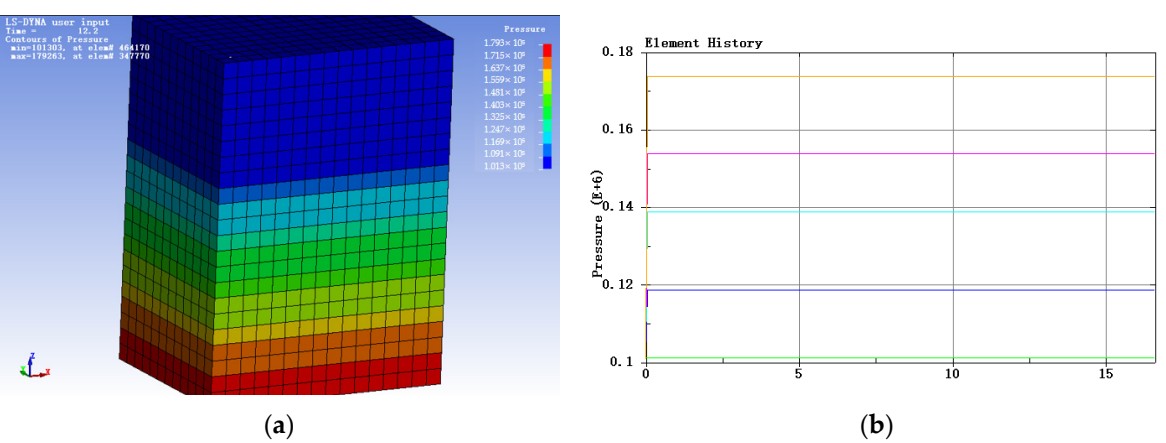

(**a**)                                                                                     (**b**)

**Figure 4.** (**a**) The distribution of pressure gradient; (**b**) the pressure at monitored nodes.

It can be seen from Figure 4 that the pressure gradually increases with depth, and the pressure at the nodes, showing in color lines, remains stable. A floating ice floe is placed within the region, and the results animation is observed and the time history of the ice floe's acceleration is shown in Figure 5. As shown in Figure 5, the acceleration of the individual ice gradually stabilizes around zero, indicating floating stability.

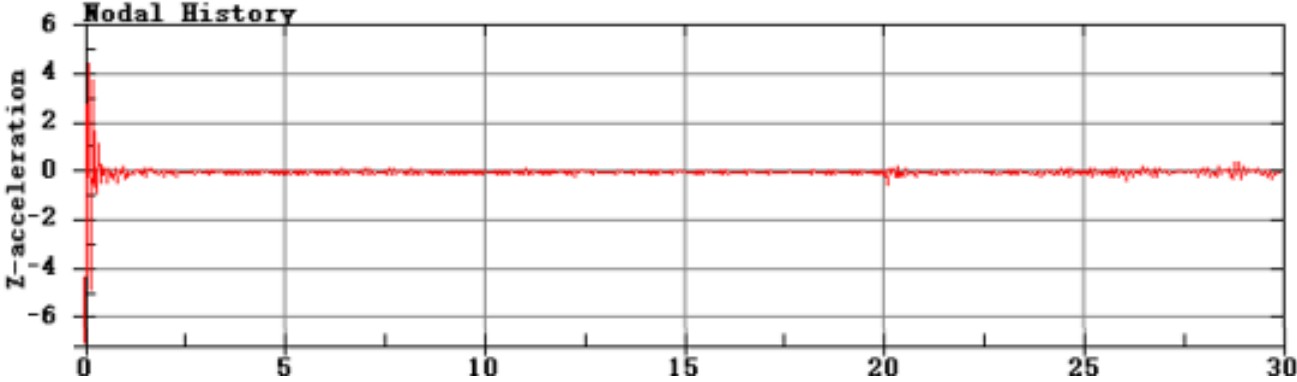

**Figure 5.** The time history of the ice floe's acceleration.

### 3.3.2. The Verification for Ice Resistance

There are four different navigation speeds considered in this research, i.e., 1 knot, 2 knots, 4 knots, and 6 knots. In the numerical calculation, a 120-core parallel computation is employed, and the 1 kn working condition requires approximately 36 h. The simulation results provide ice resistance data, excluding the open water resistance. The specific data is shown in Table 7.

**Table 7.** The numerical results based on structure-ice interaction.

| Navigation Speed (kn) | 1 | 2 | 4 | 6 |
|---|---|---|---|---|
| Maximum ice resistance (N) | $2.44 \times 10^5$ | $4.00 \times 10^5$ | $4.01 \times 10^5$ | $6.54 \times 10^5$ |

By substituting the average ice resistance into Equations (10) and (11), $\ln Fr_p$ and $\ln Cp$ can be obtained, as shown in Table 8.

**Table 8.** $\ln Fr_p$ and $\ln Cp$ data.

| Navigation Speed (kn) | 1 | 2 | 4 | 6 |
|---|---|---|---|---|
| $\ln Cp$ | 3.679 | 2.787 | 1.403 | 1.081 |
| $\ln Fr_p$ | −1.551 | −0.858 | −0.165 | 0.241 |

The relation between $Fr_p$ and $\ln Cp$ is presented in Figure 6. Based on the least squares method, the trendline equation is derived as $\ln Cp = -1.526\ln Fr_p + 1.3474$. Applying the operating conditions of a velocity of 6 knots, ice thickness of 1.2 m, and the dimensions of the initial ice floe as 6 m × 6 m, the ice Froude number is obtained. Then, the natural logarithm of $Cp$ is obtained. Finally, the theoretical ice resistance is calculated, resulting in a value of $5.91 \times 10^5$ N. In comparison, the simulation yields a predicted resistance of $6.54 \times 10^5$ N, indicating a discrepancy of 10.65%. Therefore, the obtained results are deemed reliable.

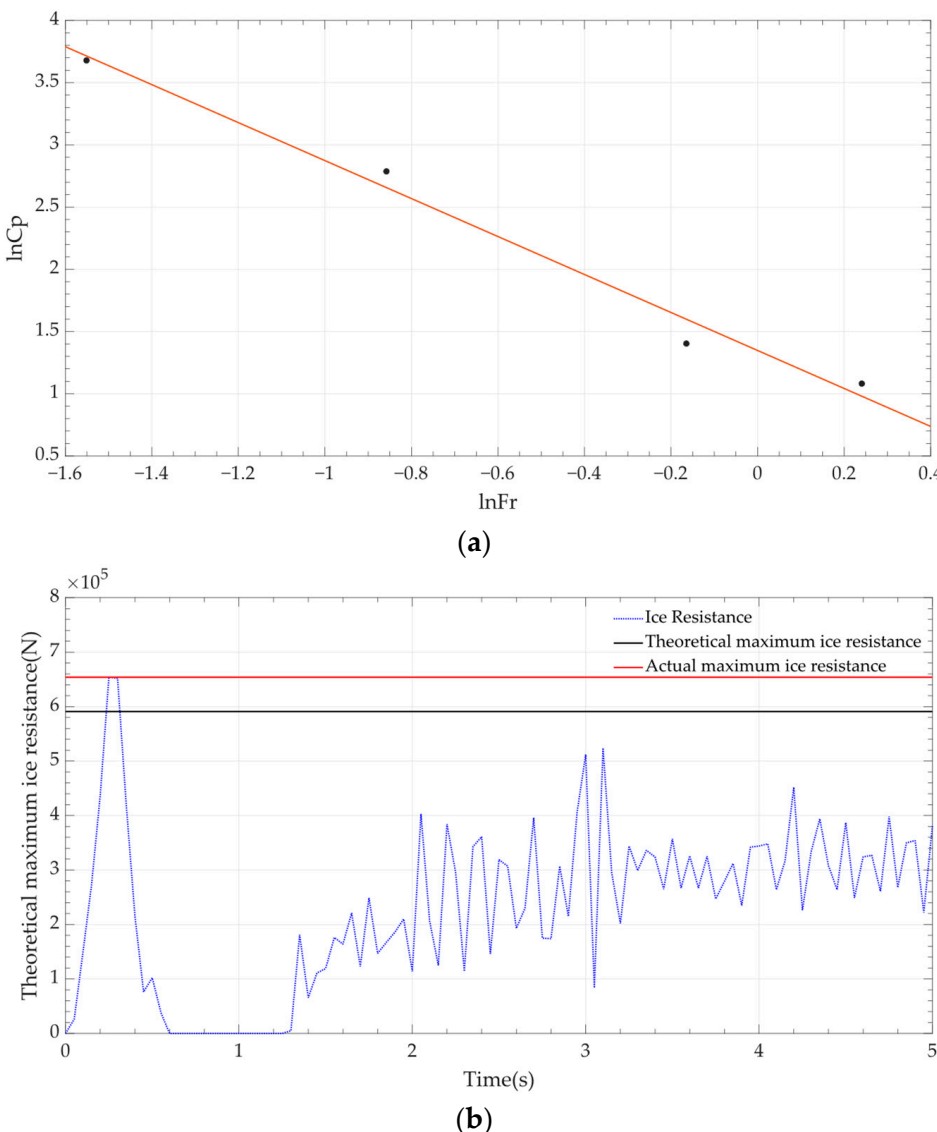

**Figure 6.** (**a**) The relation between $\ln Fr_p$ and $\ln Cp$; (**b**) the comparison of ice resistance between theory and numerical simulation.

## 4. Results and Discussion

### 4.1. The Structural Response of Structure-Ice Interaction

The work conditions in the previous section are subjected to in-depth analysis. Combined with the animation and time history curve, some mechanism of submarine collisions with ice floes was discovered. Different from traditional icebreakers due to the similar size between the submarine and the ice floes, the submarine almost only collides with the first ice floe, and the remaining floes are pushed aside without accumulating. Following impact, the ice floe momentarily attains higher velocity than the structure due to the latter's significantly greater mass, subsequently decelerating until the next impact occurs, as shown in Figure 7. This repetitive loading–unloading process leads to multiple peaks in the response. The location where the structure encounters collisions and the internal circular plate experience significant stress, making them susceptible to plastic deformation.

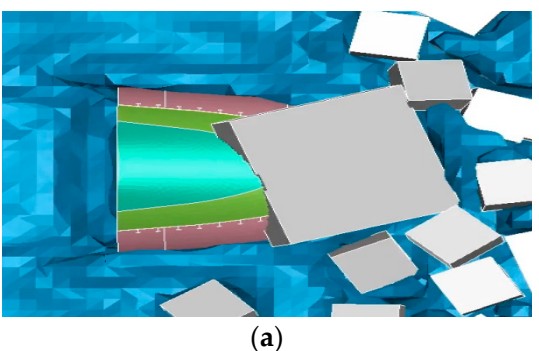

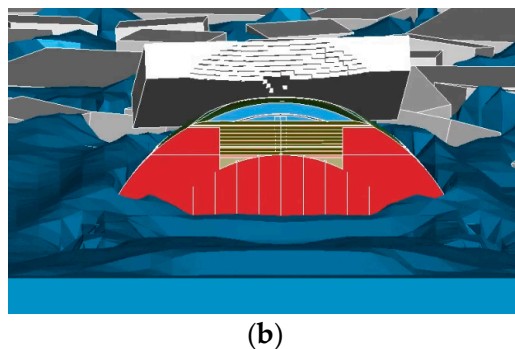

**Figure 7.** (**a**) loading stage; (**b**) unloading stage.

*4.2. The Parametric Study of Structure-Ice Interaction*

4.2.1. The Effect of Ship Speed on Structural Response

To study the effect of navigation speed on ice resistance, the ice thickness is maintained at 1.2 m, and the dimensions of the initial ice floe remain constant at 6 m × 6 m. The results are computed for velocities of 6 knots, 4 knots, 2 knots, and 1 knot, respectively. The corresponding results are shown in Figures 8 and 9.

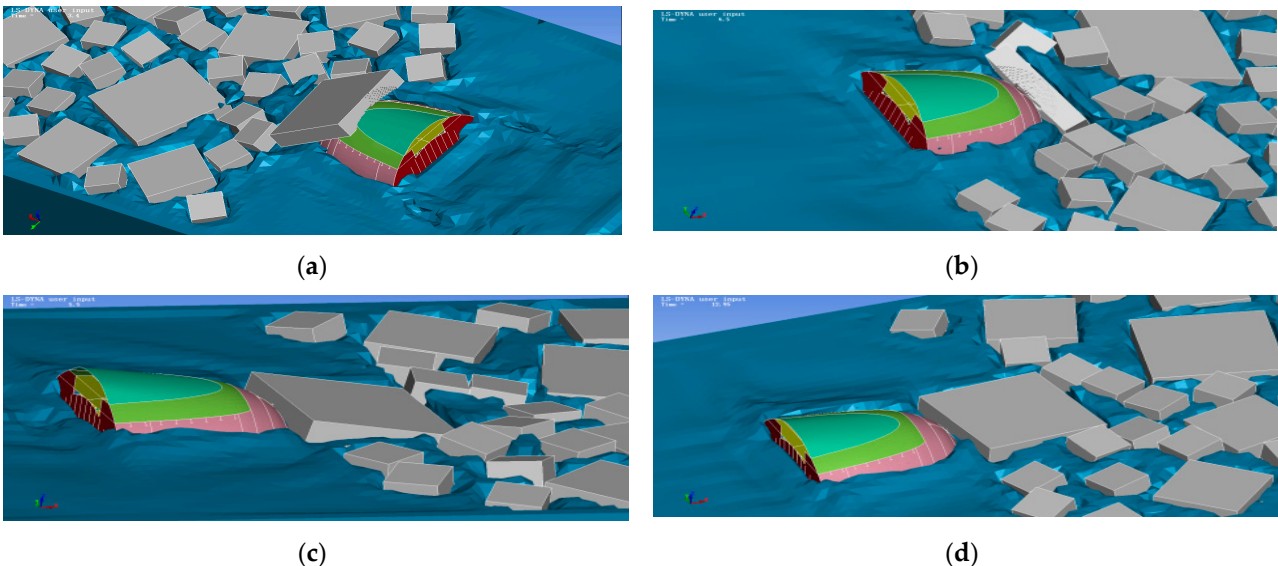

**Figure 8.** (**a**) The structure-ice interaction at 6 kn; (**b**) the structure-ice interaction at 4 kn; (**c**) the structure-ice interaction at 2 kn; (**d**) the structure-ice interaction at 1 kn.

It can be seen from Figure 8 that when the structure is moving at a speed of 1 knot, a small amount of fragmentation occurs in the first ice floe, and it is predominantly translating along the x-direction. As the submarines move forward, the ice resistance acting on the structure gradually decreases. When the structure sails at speeds of 2 knots, 4 knots, and 6 knots the arc surface at the head of the structure causes the ice floe to turn over, thereby affecting the load characteristics. The maximum and average values of the ice resistance over time are extracted and listed in Table 9.

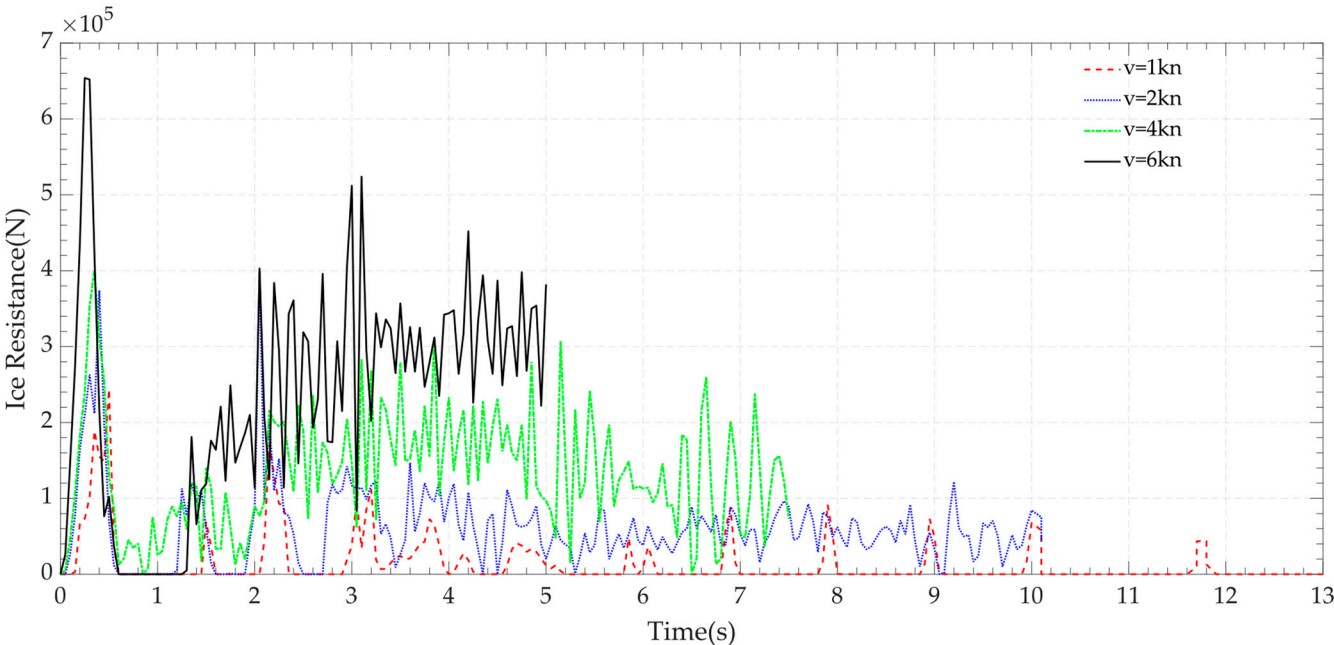

**Figure 9.** The ice resistance at different navigation speeds.

With the increase in navigation speed, both the average and maximum ice resistance increase. Specifically, there is a slight difference in the maximum ice resistance between 4 knots and 2 knots. It can be seen from Figure 8 that there is a direct connection between the relative position of interaction between ice resistance and structural response in the floating ice field.

**Table 9.** The ice resistance at different navigation speeds.

| Navigation Speed (kn) | Average Ice Resistance (N) | Maximum Ice Resistance (N) |
|:---:|:---:|:---:|
| 1 | $1.376 \times 10^4$ | $2.44 \times 10^5$ |
| 2 | $6.154 \times 10^4$ | $4.00 \times 10^5$ |
| 4 | $1.329 \times 10^5$ | $4.01 \times 10^5$ |
| 6 | $2.306 \times 10^5$ | $6.54 \times 10^5$ |

The navigation speed can affect the motion of ice floe, especially the first floe. There is some overturning of ice floe that can be observed at 1 knot and 2 knots. When the speed exceeds 4 knots, the ice floe flips upward, reaching the top of the structure, resulting in a significantly different form of interaction compared to the previous two scenarios.

Furthermore, the effective stresses and plastic strains of the structure at different navigation speeds are shown in Figure 10.

The plastic strains occur can be found in Figure 10. Both effective stress and structural deformation increase as the vessel speed increases. Although the maximum ice resistance is similar at 2 knots and 4 knots, plastic deformation occurs when the speed exceeds 4 knots, while no plastic deformation occurs when the speed is below 2 knots. Thus, when navigating in icy waters with ice floe sizes not smaller than 6 m × 6 m and ice thicknesses not less than 1.2 m, it is recommended to maintain a navigation speed below 4 knots.

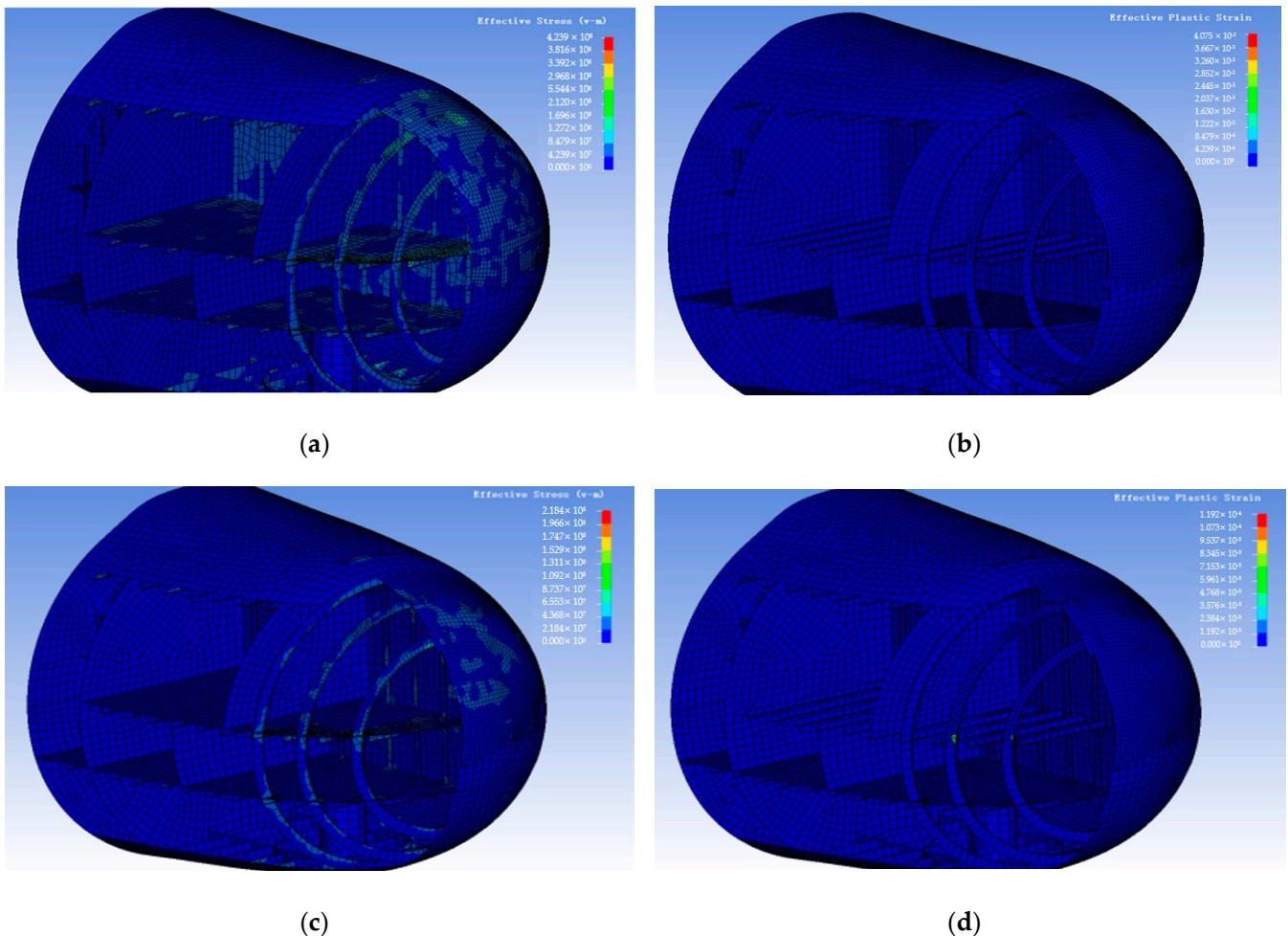

(**a**)

(**b**)

(**c**)

(**d**)

**Figure 10.** (**a**) The effective stress at 6 kn; (**b**) the structural deformation at 6 kn; (**c**) the effective stress at 4 kn; and (**d**) the structural deformation at 4 kn.

4.2.2. The Effect of Ice Floe Size on Structural Response

To study the influence of ice floe size with an ice thickness of 1.2 m and a vessel speed of 6 knots, the first ice floe is modeled in three different sizes: 6 m × 6 m, 4 m × 4 m, and 2 m × 2 m. Related results are shown in Figure 11.

It can be seen from Figures 11 and 12 that at speed of 6 knots, all three sizes of ice floes experience upturning, indicating a similar mode of motion. The unloading time is influenced by the size of the floating ice. After the collision, smaller ice floes attain higher velocities, leading to easy detachment from the structure, and the subsequent re-contact occurs later. The ice resistances with different ice floe sizes are shown in Table 10.

It can be seen from Table 10 that as the ice floe size increase, both the maximum ice resistance and average ice resistance increase. For the structural stress and plastic deformation, it is observed from Figure 13 that plastic deformation only occurs in the 6 m × 6 m size scenario, while the other two scenarios do not exhibit plastic deformation. Therefore, when the structure is sailing at a speed of 6 knots with an ice thickness of 1.2 m, collisions with ice floes of size 6 m × 6 m should be avoided as much as possible.

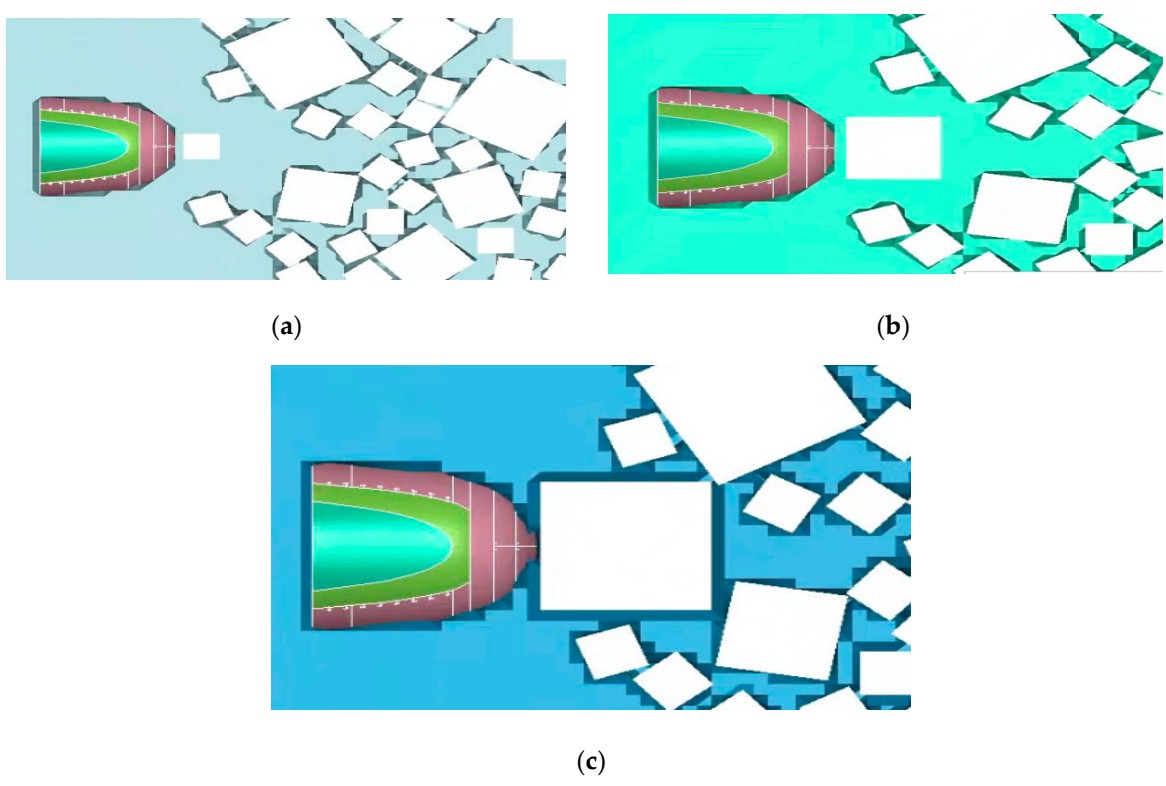

**Figure 11.** (**a**) The structure-ice interaction with size 2 m × 2 m of the ice floe; (**b**) the structure-ice interaction with size 4 m × 4 m of the ice floe; (**c**) the structure-ice interaction with size 6 m × 6 m of the ice floe.

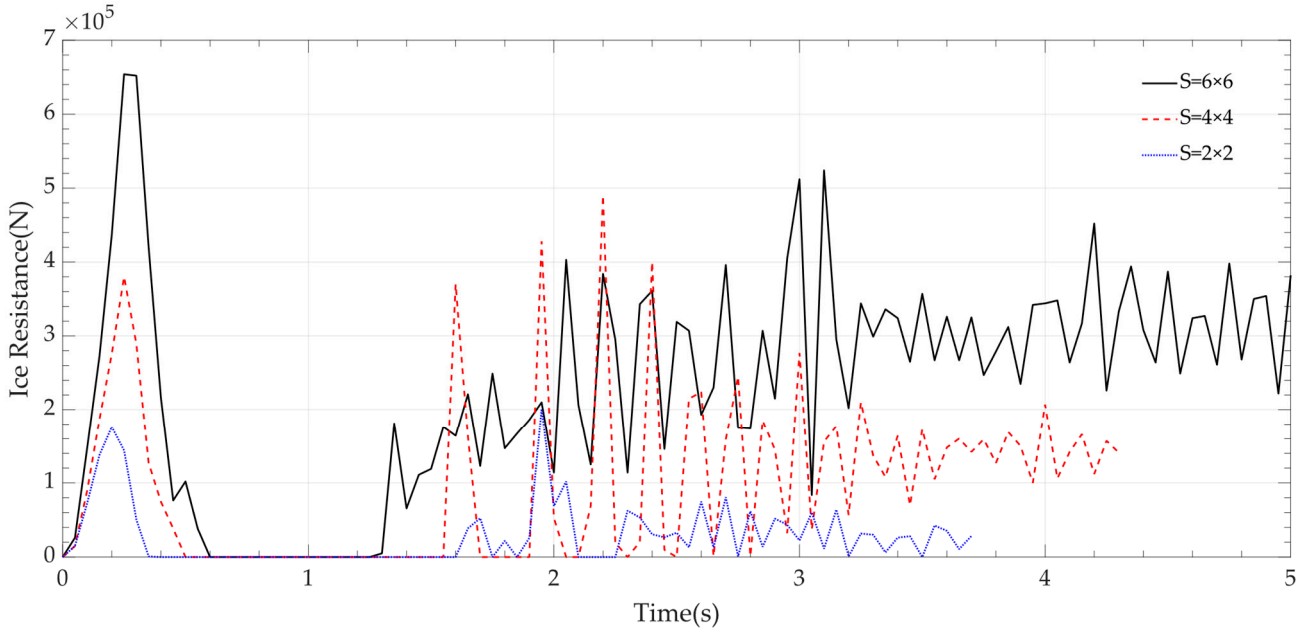

**Figure 12.** The ice resistance with different ice floe size.

**Table 10.** The ice resistances with different ice floe size.

| Ice Floe Size (m × m) | 2 × 2 | 4 × 4 | 6 × 6 |
|---|---|---|---|
| Maximum ice resistance (N) | $2.01 \times 10^5$ | $4.9 \times 10^5$ | $6.54 \times 10^5$ |
| Average ice resistance (N) | $2.755 \times 10^4$ | $9.845 \times 10^4$ | $2.306 \times 10^5$ |

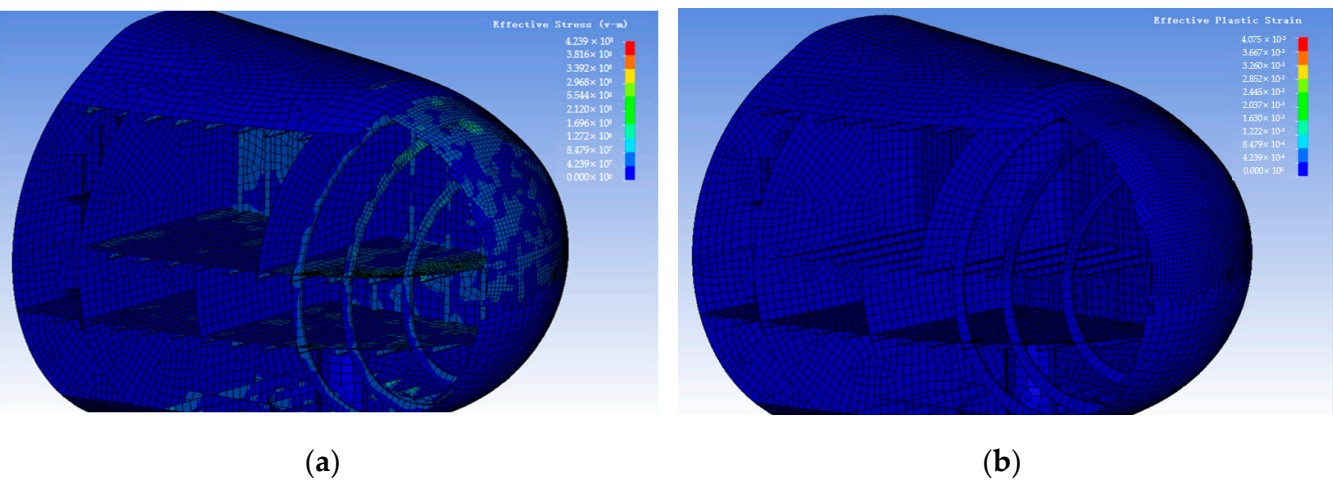

(**a**)                                                    (**b**)

**Figure 13.** (**a**) The effective stress at 6 m × 6 m; (**b**) the structural deformation at 6 m × 6 m.

### 4.2.3. The Effect of Ice Thickness on Structural Response

When studying the influence of ice thickness, while maintaining a vessel speed of 6 knots and a first ice floe size of 6 m × 6 m, the results are calculated for ice thicknesses of 0.5 m, 0.8 m, 1.0 m, and 1.2 m, as shown in Figures 14 and 15.

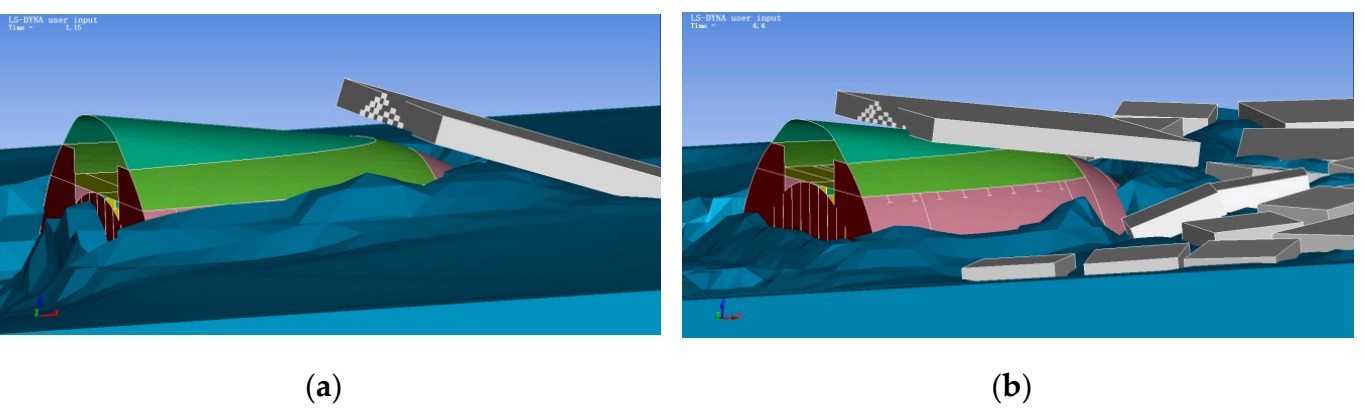

(**a**)                                                    (**b**)

**Figure 14.** (**a**) The result animation of 0.5 m working condition t = 1.15 s; (**b**) the resulting animation of 0.5 m working condition t = 4.40 s.

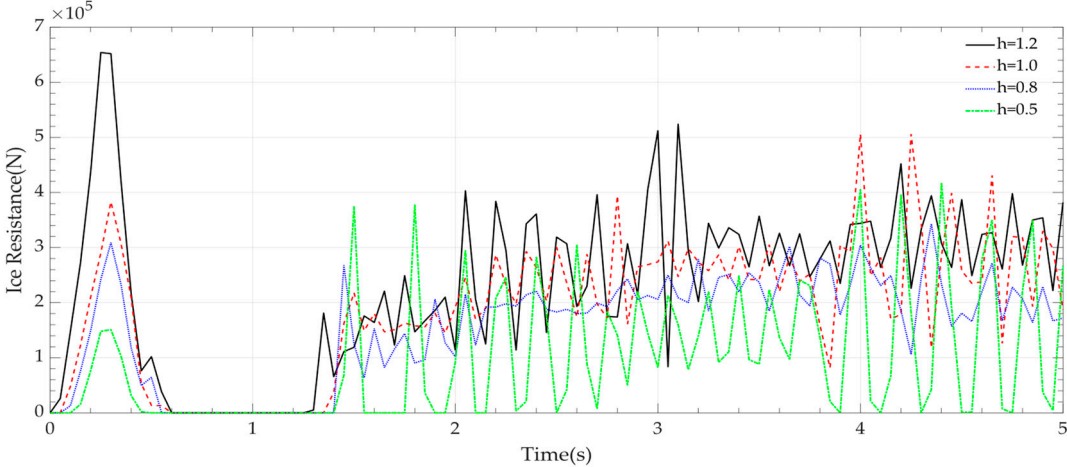

**Figure 15.** The ice resistance at different ice thickness.

The process of structure-ice interaction with thicknesses of 0.8 m, 1.0 m, and 1.2 m is similar. However, when the ice thickness is 0.5 m, the ice floes are light and easily flip to the top of the structure, undergoing multiple impacts and rebounds. Combined with the analysis of Figure 15, the influence of ice thickness is similar to that of ice floe size, whereby a smaller ice thickness results in a longer duration of separation from contact after impact. The extracted the ice resistance data are shown in Table 11.

**Table 11.** The ice resistance at different ice thickness.

| Ice Thickness (m) | Average Ice Resistance (N) | Maximum Ice Resistance (N) |
|:---:|:---:|:---:|
| 0.5 | $9.922 \times 10^4$ | $4.18 \times 10^5$ |
| 0.8 | $1.555 \times 10^5$ | $3.43 \times 10^5$ |
| 1.0 | $1.913 \times 10^5$ | $5.06 \times 10^5$ |
| 1.2 | $2.306 \times 10^5$ | $6.54 \times 10^5$ |

The structure-ice interaction at the four ice thicknesses shows similar behavior. When the ice thickness increases, both the maximum ice resistance and average ice resistance gradually increase. By extracting the structural stress and plastic deformation data, it is observed from Figure 16 that plastic deformation only occurs when the ice thickness is 1.2 m, while the other three scenarios do not exhibit plastic deformation. Therefore, when the structure is sailing at a speed of 6 knots with a floe size of 6 m × 6 m, collisions with ice floes of 1.2 m thickness should be avoided as much as possible.

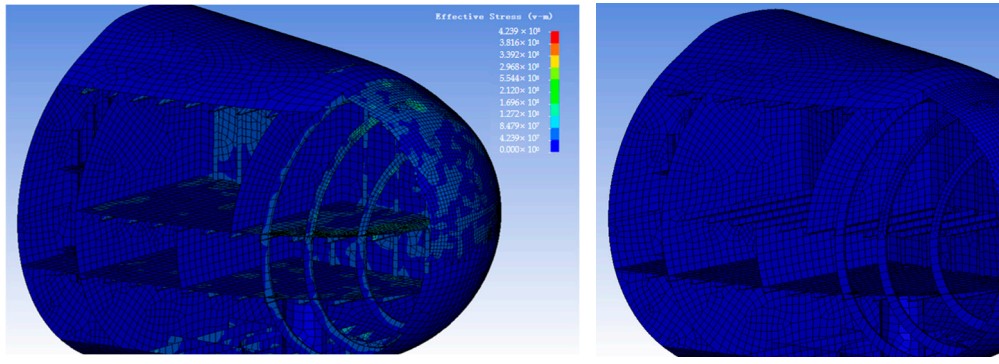

(**a**) The effective stress at 1.2 m      (**b**) The structural deformation at 1.2 m

**Figure 16.** The structural response at different ice thicknesses.

### 4.3. The Updated Ice Resistance Method

The effect of three factors (navigation speed, ice floe size, ice thickness) on the interaction between submarine and floating ice was studied in this work. The employed Colbourne method, which is only suitable for floating ice conditions, is solely dependent on navigation speed, ice thickness, and ice concentrations, disregarding the size of the floating ice, which is inconsistent with reality and contradicts the simulation results. This work attempts to incorporate the parameter *S* to represent the size of the floating ice cross-sectional area and introduces a new ice Fourier number, denoted as Equation (14).

$$Fr_p = \frac{V}{\sqrt{gh_i CS}} \tag{14}$$

where, *S*—the size of the floating ice cross-sectional area

The definition of *Cp* remains unchanged. Condition 3 represents the critical scenario of both flipped and unflipped floating ice, exhibiting a complex behavior. Condition 7 involves thinner floating ice that flips upwards and undergoes multiple loading–unloading events, where the ice in contact with the bow is not solely limited to the initial ice floe. By excluding

the data from these two particular scenarios, the results for the remaining seven conditions are calculated and plotted in Figure 17 and Table 12, yielding the following outcomes:

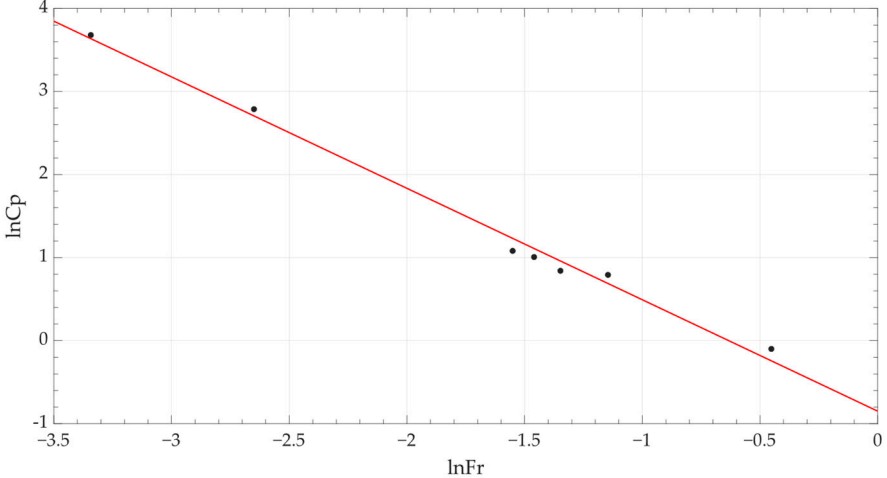

**Figure 17.** The relation between $\ln Fr_p$ and $\ln Cp$.

**Table 12.** Ice resistance results after equation improvement.

| Number | Case Condition | Ice Resistance Based on Numerical Simulation | $\ln Fr_p$ | $\ln Cp$ | Ice Resistance Based on the Updated Method | Error |
|---|---|---|---|---|---|---|
| 1 | 6 kn, 6 × 6, 1.2 m | $6.54 \times 10^5$ N | −1.551 | 1.081 | $7.60 \times 10^5$ N | 13.99% |
| 2 | 1 kn, 6 × 6, 1.2 m | $2.44 \times 10^5$ N | −3.343 | 3.679 | $2.33 \times 10^5$ N | −4.42% |
| 3 | 2 kn, 6 × 6, 1.2 m | $4.00 \times 10^5$ N | −2.650 | 2.787 | $3.69 \times 10^5$ N | −8.45% |
| 6 | 6 kn, 6 × 6, 0.8 m | $3.43 \times 10^5$ N | −1.348 | 0.841 | $3.87 \times 10^5$ N | 11.19% |
| 7 | 6 kn, 6 × 6, 1.0 m | $5.06 \times 10^5$ N | −1.460 | 1.007 | $5.61 \times 10^5$ N | 9.76% |
| 8 | 6 kn, 2 × 2, 1.2 m | $2.01 \times 10^5$ N | −0.452 | −0.099 | $1.74 \times 10^5$ N | −15.41% |
| 9 | 6 kn, 4 × 4, 1.2 m | $4.90 \times 10^5$ N | −1.146 | 0.792 | $4.41 \times 10^5$ N | −11.02% |

Where, Error = (theoretical − simulation)/theoretical × 100%.

To verify the rationality of the updated ice resistance method, this study computes the case where the initial ice floe size is 5 m × 5 m, navigation speed is 6 knots, and ice thickness is 1.2 m. The maximum ice resistance obtained is $5.43 \times 10^5$ N, while the theoretical value calculated using the equation is $5.95 \times 10^5$ N, resulting in an error of 8.80%, which falls within an acceptable range. Based on these results, an updated ice resistance method is proposed in this work, which takes into account parameters such as navigation speed, ice thickness, and floating ice size comprehensively, making it a suitable empirical equation for this submarine.

## 5. Conclusions

In this work, the dynamic response of submarines navigating in floating ice field was conducted based on the FEM-ALE coupled method. Different from traditional icebreakers, the submarine almost only collides with the first ice floe, and the remaining floes are pushed aside without accumulating. The collision is a continuous loading–unloading process, and its frequency is negatively correlated with the three working condition factors. The intensity of dynamic response of the submarine is generally positively correlated with the three working condition factors, but it is locally influenced by the relative position of the collision. All three working condition factors affect the degree of ice floe flipping, thereby influencing the relative collision position.

In this work, plastic deformation of the structure only occurred under the conditions of an ice thickness of 1.2 m, floating ice size of 6 m by 6 m, and navigation speeds of 6 knots

and 4 knots. Therefore, it is recommended that the structure should try to avoid more severe conditions than an ice thickness of 1.2 m, a floating ice size of 6 m by 6 m, and a navigation speed of 4 knots when navigating on floating ice field. Based on the simulation results in this paper, an updated ice resistance method was developed by introducing the floating ice size to the Colbourne method, which was validated using a 5 m × 5 m condition.

Due to the extensive computational time required for each scenario in this study, the mechanism of the impact of floating ice flipping on collisions was not quantitatively investigated. The critical working conditions currently proposed are estimated values that require further refinement into more accurate values through additional research. The empirical equation improvement scheme presented in this paper, based on simulation results, needs to be validated through model tests.

**Author Contributions:** Conceptualization, Z.C. (Zhanyang Chen) and H.G.; software, Z.C. (Zhongyu Chen); writing—original draft preparation, Z.C. (Zhongyu Chen); writing—review and editing, W.Z. and H.R.; supervision, W.Z.; funding acquisition, W.Z. and G.F. All authors have read and agreed to the published version of the manuscript.

**Funding:** This paper was supported by the Shandong Provincial Natural Science Foundation, Fund No. ZR2022QE092 and the National Natural Science Foundation of China, Fund No. 52171301, China Postdoctoral Science Foundation, No. 2023M730829.

**Institutional Review Board Statement:** Not applicable.

**Informed Consent Statement:** Not applicable.

**Data Availability Statement:** Data available on request due to restrictions eg privacy or ethical. The data presented in this study are available on request from the corresponding author. The data are not publicly available due to the structure of the submarine is relatively sensitive.

**Conflicts of Interest:** The authors declare no conflict of interest.

## Nomenclature

| | |
|---|---|
| ALE | Arbitrary Lagrangian-Eulerian |
| $B$ | maximum beam for the structure |
| $C$ | the intercept of the $\mu_s - \mu_p$ curve (speed of sound in water) |
| CFD | Computational Fluid Dynamics |
| $C_{OW}$ | The open water resistance coefficient |
| $C_P$ | The drag coefficient of ice floe |
| DEM | Discrete Element Method |
| $E$ | The internal energy per initial volume |
| EOS | Equations Of State |
| FEM | Finite Element Method |
| $F^{\text{int}}(t_n)$ | Internal force vector |
| $Fr_p$ | The ice Froude number |
| $g$ | The acceleration of gravity |
| $h_i$ | The ice thickness |
| $H(t_n)$ | Hourglass resistance |
| $M$ | Mass matrix |
| $P(t)$ | External force vector |
| $R_{ow}$ | Open water resistance |
| $R_p$ | Ice resistance |
| $S$ | The size of the floating ice cross-sectional area |
| SPH | Meshless Method |
| SRC | Strain rate parameter C |
| SRP | Strain rate parameter p |
| $S_1$ | The coefficients of the slope of the $\mu_s - \mu_p$ curve |
| $V$ | The navigation speed |

| $\alpha$ | The first-order volume correction to $\gamma_0$ |
|---|---|
| $\gamma_0$ | The Gruneisen gamma parameter |
| $\rho_i$ | The density of ice |

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
