# Peer review of "Dynamic Response Analysis of Submarines Based on FEM-ALE Coupling Method in Floating Ice Conditions"

_jmse, doi:10.3390/jmse11081560_

Round 1

Reviewer 1 Report

A file with comments/corrections has been uploaded in the present App.

Some important/relevant improvements should be introduced. Sentences as the one which starts the Introductions "...the Arctic has become an area of significant development significance." seems to have been written carelessly and diregarding most Journal Quality rules. 

Reviewer 2 Report

Minor English editing is required.

Reviewer 3 Report

The authors have tried to understand the ice forces acting on a submarine when it is operating at the surface. The paper is generally well written, and uses commercial computer code combining ice elements with CFD to calculate the interactions with the ship's hull. The reviewer thinks the paper can be improved in the following areas; 

1. The paper is clearly about the hull of a submarine ship operating at the surface. As a result, the title of the paper is somewhat misleading. Underwater structures is a term that can mean equipment on the sea bed, such as the type of equipment that is common in offshore energy projects. 

2. The authors should check the context for their use of Colbourne's method. It is primarily a way for non-dimensionalizing the results of model experiments, in order to make accurate predictions of full scale forces. The approach taken was to reflect that contact forces between a ship and pack ice (at low concentrations) or icebergs was impulsive in nature, and more difficult to understand, compared to more steady state forces, such as level ice or open water forces, where a mean force is a suitable measure. The method allows for the consistent presentation of forces, and allows for a the size and shape of the floe sizes. It's primary function is to smooth out the uncertainties in model experiments, and to allow for the fact that ice density and water density at model scale are different from the full scale values. 

Colbourne's method also does not include any shape factors for the hull. While this might not be important for a single impact, it does become more important as ice concentrations increase, and ice pieces jam on each other rather than just coming into contact with the hull.  

While the use of the method to normalize the results is valid, the approach does not really contain any predictive properties beyond the original data. Therefore using it to estimate a maximum force might be beyond its capability. As a result, the prediction of maximum load might be suspect. 

Also, in Colbourne's method, B is the beam of the ship at the waterline. In the case of a submarine ship, is beam at the waterline used, or maximum beam for the ship? 

Overall the paper is well written. Some paragraphs are a bit long, and could be broken into smaller parts. 

Round 2

Reviewer 1 Report

A file with new comments has been uploaded in the App.

Minor editing of English language required

Reviewer 2 Report

No more further comments.

Minor editing of English language is required.

Reviewer 3 Report

The reviewer thanks the authors for their detailed explanation. I note that the simulations were only run for a single value of ice concentration of 50%. Colbourne's method allows for different ice concentrations, and it is hard to say that the method applies when only a single, and relatively low value of this parameter was used. I stand by my argument that Colbourne's method is based on experiment results, and is therefore empirical, but I accept the authors' clarification. 

The paper can be strengthened by some more detailed modification to the English. 

Line 9, word missing?, few research studies? 

Line 22 'significant development significance' is awkward phrasing. 

Line 58, after reference [15] the format of the citation changes. Until then just family names had been used for authors. After that sometimes names and sometimes initials are also used with the family name. Format should be consistent throughout the paper. 

Line 72, check spelling of Mttnen

Nomenclature, B (beam of ship) is missing from the list of variables. 
